# The Mediating Roles of Economic, Socio-Cultural, and Environmental Factors to Predict Tourism Market Development by Means of Regenerative Travel: An Infrastructural Perspective of China–Pakistan Economic Corridor (CPEC)

**Maria Zulfaqar [1], Shahid Bashir [2,\*], Samer Mohammed Ahmed Yaghmour [3], Jamshid Ali Turi [4] and Musaib Hussain [1]**

1    Department of Management Science, National University of Modern Languages, Islamabad 44000, Pakistan
2    Business School, Tecnológico de Monterrey, Monterrey 64700, Mexico
3    Department of Travel and Tourism, Faculty of Tourism, King Abdulaziz University, P.O. Box 80207, Jeddah 21589, Saudi Arabia
4    College of Business Administration, University of Tabuk, Tabuk 71491, Saudi Arabia
\*    Correspondence: shahid.bashir@tec.mx

**Abstract:** Even though the significance of the China–Pakistan economic corridor (CPEC) is frequently discussed on various international forums, its economic, socio-cultural, and environmental impacts in a geographically constrained area have not yet been studied precisely. Consequently, the goal of this study is to look into how CPEC Infrastructural Development (CPECID) would regenerate the tourism market in Gilgit Baltistan (GB), a Pakistani administrative territory. The basic data gathered via a convenience sample strategy is subjected to a quantitative analysis approach. In total, 336 inhabitants of GB participated in a closed-ended online survey that was used to gather data. The results showed that CPECID has a favorable influence on regenerative tourist growth and development in the area and that this link is partially mediated by economic, socio-cultural, and environmental impacts. The study's conclusions have important implications for authorities creating regenerative tourist promotion plans, in addition to adding to the body of knowledge on tourism.

**Keywords:** regenerative travel; purposeful travel; responsible travel; China–Pakistan economic corridor (CPEC); tourism and travel

## 1. Introduction

Infrastructure development is essential for tourism growth and sustainability [1]. A solid road infrastructure attracts a significant number of visitors [2–4]. The upgrading of roads adds to the expansion of the tourism sector [5–7]. Infrastructure development enhances people's quality of life [8–10], increases tourism attractions, and draws more tourists [11,12]. However, a few studies have found that, even though road infrastructure in this area promotes modern tourist attractions, it has minimal effect on existing tourism activities (e.g., regenerative tourism) [13,14].

Pakistan's biological richness and natural beauty have historically considerably increased tourism demand in this region. The greatest mountain ranges (Himalaya and Hindukush), resorts, lakes, glaciers, and other magnificent natural regions attract visitors from across the world [15,16]. Previous researchers discovered that road transportation modes such as rail, road, and air had a good influence on tourist activities [17,18]. According to [10], tourists from industrialized nations are mostly addicted to utilizing contemporary transportation infrastructure. Tourists, by the means of regenerative travel, utilize the road and other modes of transportation to get to their destinations. Therefore, making and improving better road and transportation infrastructure decreases travel time and expenses [5]. If the preferred tourist destination has bad road and transit infrastructure

and terrible travel, the prospective visitor would regenerate alternate tourist destinations. As a result, improving roads and transportation is recommended as a required prediction for tourist operations [10,15]. Thus, keeping in view the relationship between infrastructure and regenerative tourism development, this study examines the impact of CPEC infrastructural development and its impact on the tourism industry in Pakistan [15,19,20].

Over seven decades, China and Pakistan have maintained a friendly and mutually beneficial relationship. In recent years, the great relationship between the two nations has led to the development of an agreement that will be very advantageous not just to both parties but to the whole globe. The China–Pakistan economic corridor (CPEC) is the centerpiece project of the $5 trillion one belt, one road (OBOR) investment project [21] initiated by China. CPEC consists of constructing an economic corridor that promotes the bilateral connection, construction, and exploration of possible bilateral investment, economic and trade, logistics, and people-to-people interactions for regional connectivity in order to better the lives of Pakistani and Chinese citizens. It was presumed that the project would offer a significant economic incentive for Pakistan's government and attract further foreign investments [22], benefitting not just Chinese investors but all foreign investors (i.e., multinational firms), including those from East Asia, the United States, and the European Union [18,22,23].

The CPEC project has considerable public support and ownership among Pakistanis, particularly in terms of boosting the local tourism sector, socioeconomic development, poverty reduction, and improving living circumstances in a variety of geographically located areas [24–26]. It is anticipated that this initiative will have a good effect on economic cooperation on a worldwide scale. In contrast, the scientific community is concerned about the potential economic, social, and environmental impacts of this undertaking [9,27,28]. Furthermore, opponents contend that CPEC is insufficient to provide Pakistan with more than moderate economic development advantages since the project's benefits are believed to exceed its costs [29,30] particularly in terms of having a minimal impact on regenerating tourism activities [14].

Due to the proximity of Gilgit Baltistan (GB) (an administrative territory of Pakistan) to the CPEC, the success of the latter will have far-reaching effects on the local economy. Since the CPEC will connect the region to the port of Gawadar, GB will become a worldwide economic hub [31]. This will assist in boosting the trade of local minerals and gems, such as emerald, sapphire, ruby, aquamarine, moonstone, and amethyst; the CPEC will also connect the region to the port of Karachi [31]. Agriculture is a big percentage of the primary source of nourishment for the whole population, and the region produces around 100,000 metric tons of organic apricots, 4000 tons of organic cherries, and 20,000 tons of organic apples, according to the Asian Development Bank (ADB); the people expect that by exporting these fruits to China, their sales and earnings will increase [32].

Keeping in mind the foregoing discussion, as well as the fact that while road infrastructure in an area generates modern tourist attractions through reflections of economic, social-cultural, and environmental impacts, it has little effect on current tourism activities [14], the aim of this study is to investigate the impact of CPEC road infrastructure on regenerative tourism development through the mediating roles of economic benefit, social-cultural, and environmental impacts. The research will focus only on the GB market sector since the success of the GB is projected to have far-reaching consequences for the local economy.

As a result, the emphasis remains on the following primary research objectives:

1.  Research the effect of CPEC infrastructure development (CPECID) on the GB tourist economy.
2.  Investigate the economic, socio-cultural, and environmental advantages and costs associated with CPECID and tourism in GB.

## 1.1. Global Market Perspective

GB, an administrative territory of Pakistan, has 1.5 million inhabitants and 27,188 square miles (72,000 square kilometers); GB comprises three divisions—Chilas, Gilgit, and Baltistan—and 10 districts [9]. The Wakhan strips (Afghanistan) and Xinjiang province of China border the north, while Indian Occupied Kashmir borders the south and southeast and Chitral borders the west (KPK) (see Figure 1). Five of the fourteen highest mountains in the world are in this region. GB's beautiful glaciers feed rivers (River Indus) that store 75% of its water. GB has mountains, Deosai plateau, lakes, the Karakoram Highway, glaciers, deserts, forests, flora, fauna, heritage, culture, and traditions. In 2017, 1.72 million people visited GB; this number is expected to increase to around 2.5 million this year [10]. Well-planned tourism may make money. The CPEC gateway project relies on Gilgit Baltistan. It is the entryway to CPEC and is crucial to its success.

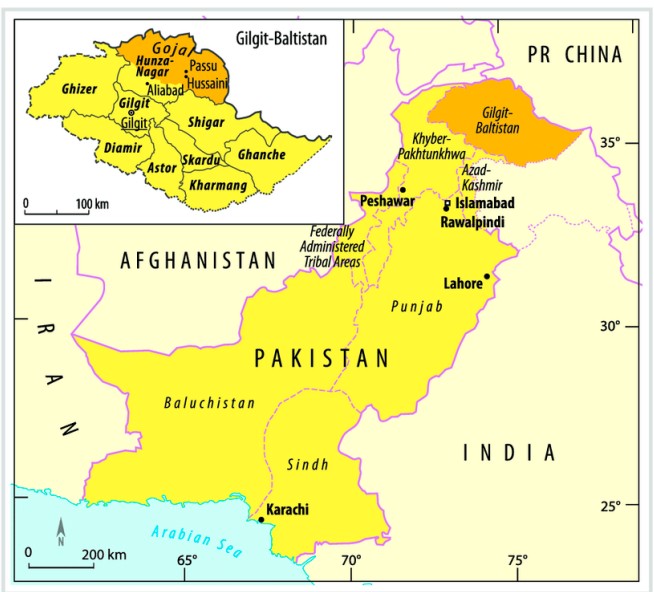

**Figure 1.** Map of Gilgit Baltistan [28].

In GB, tourist activities regenerated significantly with the completion of the Silk Route in the late 1970s, yet the area is still in the early phases of regenerative tourism development owing to a lack of tourism infrastructure. An increase in tourism activity was expected in the region after the completion of the CPECID project [9]. Therefore, this region is selected as the subject of this study because of its natural beauty, hospitality services, and the regenerative development of the CPEC infrastructure.

Modern researchers often advocate CPEC investments since they believe they will benefit the global economy the greatest [9,18,22,23,26–28,30,33]. As a result, the importance of CPEC was emphasized, and it was acknowledged that this project has the potential to be a global game-changer, especially in the regenerative tourism market. CPEC will convert and enhance Pakistan's economy from a poor level of performance to a high level of accomplishment, transforming Pakistan into a commercial center and facilitating commerce with nations in Central Asia, Europe, and the United States, according to many studies [18,22,23]. Moreover, the CPEC project is becoming a glimmer of hope for future cooperation not just between Pakistan and China but also among economically significant nations such as Afghanistan, Iran, and Turkey, who are all interested in joining in this project [22]. The creation of economic cooperation between these nations would aid in resolving a range of issues, particularly those pertaining to tourism, and increase economic stability in general.

The research framework and the relevant hypotheses are included in the discussion of the appropriate literature review, which follows the study's introduction. The reader

will next be given the findings, analysis, and discussions. We will talk about the implications of this discovery for theory and practice in the next section. The section that concludes the whole study discusses the study's shortcomings as well as potential areas for further investigation.

### 1.2. Regenerative Travel Perspective

Ref. [34] defines "regenerative" as "providing the circumstances for life to continually renew itself, to transcend into new forms, and to prosper despite ever-changing living situations". Ref. [35] introduced "regenerative" to tourism. Regenerative tourism creates circumstances for the sector to rebirth and evolves without much human involvement [36–38]. Regenerative tourism recognizes that tourists and destinations are part of a living system in nature and follow its guidelines; it recognizes the connection of natural and social surroundings and gives back to the earth and people [37,39]. Since tourism is not a stand-alone business, this transformative word presents significant challenges. The key issue will be how much human intervention, organizational, and service growth may interfere with life's logic. Destination management, marketing, and product development will disrupt natural tourism recovery, making regenerative tourism difficult [37].

Regenerative tourism goes above and beyond sustainable tourism by emphasizing "giving back" and actively regenerating local people, cultures, histories, places, and landscapes [39–43]. Having a regenerative mentality requires shifting from an individualistic and egocentric viewpoint to one that emphasizes ecology and collaboration. To enhance the performance of social and natural systems, the Global Initiative for Regenerative Tourism emphasizes changing how a person interacts with themselves, other people, and non-humans [39].

The global travel renaissance may not have fully happened yet, but the rising numbers of travelers who have had all of the recommended vaccines are observed to travel during the fading new COVID-19 variants. By 2023, international tourism should recover to pre-pandemic levels. However, practitioners and researchers are exploring regeneration approaches to improve tourism. Therefore, this study has been conducted to substantially add to the scarce literature and empirical data on regenerative travel and tourism and to improve the body of knowledge on sustainable tourism.

Since Pakistan and China value economic stability and environmental sustainability, this research emphasizes the relevance of international investment on regenerative tourism. During the COVID-19 pandemic, they had to follow global environmental protection standards [18]. The COVID-19 pandemic has exposed management failures and collective initiatives to repair tourist sites and over-traveled areas. Sustainable tourism is needed to regenerate GB's natural environment. This research will encourage regenerative policies and behaviors as we move toward normalcy to make travel more useful than harmful.

Finally, it is hoped that this research will contribute to the establishment of reputable and well-established market leaders in the field of regenerative tourism who have a sound physical infrastructure. It is envisaged that regenerative concepts and recovery plans will be put out because of this research. The purpose of these plans is to strategically draw contributions from the tourism sector, the benefits of which outweigh the resources they utilize. To leave GB in a better shape than we found it, we want to provide a hand in the transition from sustainable to regenerative travel habits.

## 2. Literature Review

### 2.1. Social Exchange Theory

According to social exchange theory (SET), human relationships are based upon cost–benefit analysis. If people perceive that the benefits are more than the cost, they further participate in the activity [44]. Residents are more willing to participate in a trade with visitors to their town if they have a fair prospect of coming out ahead while avoiding intolerable losses. Residents are more inclined to support future tourism development

and engage in regenerative tourism-related activities if they perceive that the prospective advantages of tourist expansion exceed any possible negatives [45].

Social exchange theory suggests that in the context of regenerative tourism, locals are more likely to participate in exchange that supports tourist development if the perceived benefits of tourism surpass the perceived costs of tourism [7,46]. People in the host community are more eager to engage in an interaction with guests if they believe it will be advantageous to them in some manner [47,48]. Research on regenerative tourism has shown that inhabitants' opinions of tourism and their desire to participate in an exchange are influenced by a variety of variables, including economic, sociocultural, and environmental (support for or opposition to tourism development) variables. In general, the interactions that take place between guests and hosts, as well as those that take place between hosts and other guests, influence the social consequences of tourism and may affect those repercussions [49]. As a result, social exchange theory will play the role of this investigation's guiding theoretical underpinning.

In a way, regenerative tourism is reliant on the hospitality of the people who live nearby. Therefore, the support of these residents is critical to the industry's development, profitability, and long-term survival. Locals who have a positive opinion of regenerative tourism are more inclined to support the growth of tourism-related infrastructure and, consequently, are more willing to engage in dialogue with visitors. On the other hand, if people believe that the expansion of regenerative tourism will result in more costs than benefits, they would most likely be hostile to its development. There is a danger that a regenerative tourism development project will fail if it is conceived of and implemented without the knowledge and support of the local community. As the breadth of the development project expands, so does the danger. The visitors will soon feel the hostility, apathy, or mistrust that is directed towards them. The great majority of regenerative tourists will avoid visiting places where they do not feel they will be welcomed. Consequently, it is critical to understand how locals assess the factors that influence the perceived overall impact of regenerative tourism development.

### 2.2. Infrastructure Development

The industry includes both the supply of regenerative tourist services and the staging of related events. If a tourist is dissatisfied with one aspect of their vacation, their whole perception of the trip suffers [50,51]. Transport and road infrastructure are critical to the increase in tourist-related activities and the general enhancement of regenerative tourism. The infrastructure services, highways, motels, hotels, and tourist resorts in the destination are critical to the economic growth of tourism [52,53]. Improvements to transportation infrastructure are expected to boost market accessibility and cut travel costs, making destinations more likely to be considered by tourists when deciding where to visit [54].

The effective growth of regenerative tourism is mainly reliant on sufficient transportation. Due to the availability of adequate transportation, inactive tourist hotspots have been converted into vibrant tourist hotspots and profitable destinations that attract large people. The construction of a tourist site should begin with the installation of suitable transportation facilities. People's impressions of the repercussions of tourism are highly impacted by the development of the underlying road infrastructure, which is crucial in shaping local attitudes about tourist growth [9]. The resulting hypothesis is stated below after the conclusion of the literature review.

**H1.** *There is a positive relationship between CPEC Infrastructural development and tourism development.*

### 2.3. Economic Impact (Benefit and Cost)

According to several research studies (i.e., [32,55–57]), the development of new roads and other kinds of transportation infrastructure creates jobs and boosts the economic activity of the surrounding community (improved local infrastructure leads to higher

affluence for the people, more and better employment opportunities, and increased demand for locally produced items). Regenerative tourism is seen positively by residents since it improves socioeconomic situations, elevates the prestige of underrepresented regions, and raises living standards for visitors. As a result, inhabitants believe tourism has a good influence on their city [8,58,59]. The development of local area infrastructure and subsequent interactions with visitors has a direct impact on communities' perceptions of the repercussions of tourism, particularly among those who gain the most economically from tourism [59–62].

The growth of a location via the installation of supporting infrastructure to regenerate further economic activity may have either good or negative consequences. Although there are several financial benefits to tourism, there are also numerous expenditures involved with it [63]. Regenerative tourist activities have good social, political, and environmental implications. The first common expense linked with regenerative tourism is the enormous sums of money spent by governments on the construction of tourist-related infrastructure. Infrastructure building, which may include roads, transportation, hotels, beaches, and other structures, exerts a substantial financial load on governments [64]. The local governments and the community's support are critical for the success of the local regenerative tourism business. Residents whose financial well-being benefits from the presence of visitors are more optimistic about the advantages of tourism and are more willing to promote tourism in their community [25,65,66]. The residents' feelings about the effects of visitors are a crucial component in deciding whether locals support or oppose the increase in tourism in a certain location; however, this varies across developed and developing nations [59]. If residents think the advantages of regenerative tourism exceed the costs, they are more likely to trade and support the expansion of tourism in their town [59–61,65].

Residents' perceptions of the good benefits of tourism inspire and urge them to support the growth of tourism in their town, but residents 'perceptions of the negative consequences of tourism discourage them from doing so [66,67]. Residents that suffer negative outcomes from tourism are less likely to participate in commerce and are less likely to support tourist growth [49,59–61]. People see traffic congestion, overcrowding, increased trash, increased crime, drugs, and prostitution as expenses connected with tourism, according to [65,68,69]. These factors also operate as roadblocks to people's willingness to support visitors. Respondents who are more financially secure, younger, and more educated are less optimistic about regenerative tourism. As a result, further study has shown that people's unfavorable impressions of the repercussions of tourism are connected to their reluctance to support the expansion of the tourism business [59,62,67,70,71]. The relationship between tourism and general economic development in the region should be the major focus of future study [72]. As a consequence of this, and after considering the relevant previous research, the following hypotheses are proposed:

**H2.** *There is a positive relationship between CPEC infrastructural development and economic benefits.*

**H3.** *Economic benefit has a positive relationship with tourism development.*

*2.4. Socio-Cultural Impact (Benefit and Cost)*

It is often assumed that earning financial rewards would enhance people's quality of life, and it is also feasible that social components may be beneficial [73]. Academics believe that the rise of tourism affects areas of socio-cultural life such as routines, beliefs, and values, in addition to habits and daily activities [74]. Furthermore, communities that attract many visitors tend to witness a growth in population as a result of new individuals migrating there from adjacent regions [75].

Regenerative tourism can provide significant social benefits, such as the promotion of community pride, the recognition of indigenous peoples, the cultivation of tolerance and acceptance through travel, the expansion of amenities (such as parks and recreational

facilities), investment in arts and culture, learning about different cultures that can be gained through travel, and the expansion of amenities. When tourism is handled well, it has the potential to and does improve the quality of life of locals while simultaneously increasing tourist knowledge, comprehension, and appreciation [76]. Tourism benefits both pride in one's town and active participation, allows tourists and locals to share their cultural experiences, encourages the preservation and enjoyment of regional festivals and cultural activities, and encourages the development of current abilities as well as the acquisition of new ones [77].

Tourism is often seen as a catalyst for positive change because of its favorable impacts on socioeconomic circumstances, greater awareness for previously hidden locales, and enhanced quality of life for visitors [8,58,78]. Furthermore, regenerative travelers from other nations contribute to the enrichment of the cultural environment of the country to which they travel. Travel allows individuals to attain better knowledge of various cultures. Furthermore, it aids young businesses in the area in producing goods and services that they would be unable to market if they were just reliant on their local community [59].

According to [79], tourism has a number of negative consequences, including the loss of traditions, higher levels of consumerism, rising crime rates, social instability, and greater levels of traffic congestion. Vacationers' hate is one of the most severe reactions, which may be compounded by factors such as growing crime rates and higher levels of traffic congestion [61]. According to [65], the negative economic effects of tourism, such as rising living expenses, product prices, and real estate prices, have a direct influence on communities that serve as tourist attractions.

A study conducted by Barcelona Field Studies Centre (2018) in Hawaii found that tourism leads to the degradation of social, economic, and environmental circumstances. According to the findings of this study, 37% of respondents believe that Hawaii's tourism business is to blame for the state's growing crime rate. Most respondents (64 percent) feel that more tourism is to blame for increased prostitution but not for increased vandalism (43 percent) or increased drug usage (27 percent). Despite the dubious link between regenerative tourism and criminal behavior, more than two-thirds of respondents believe that more money should be spent on preventing crime rather than expanding tourism. Visitors' use of drugs and alcohol can change the residents' conduct as well as the resident's relationships with their friends and family. There is a chance that the number of sexually transmitted infections may continue to climb. Traditional values and culture are disappearing as a result of individuals imitating tourist behavior or getting cultural transmission as a direct result of frequent and everyday engagement. It is probable that rich visitors will be hounded, that crime rates will rise generally, and that human rights will be infringed [77]. Future studies should look at the impact of CPECID on residents' views of regenerative tourism's effects and their support for the industry's growth in terms of socio-cultural and environmental aspects [9]. As a result of the review of the relevant literature, the following hypotheses are advanced:

**H4.** *CPECID has a social-cultural impact.*

**H5.** *Social-cultural has an impact on tourism development.*

### 2.5. Environmental Impact (Benefit and Cost)

The regenerative tourist business not only assures the preservation of natural resources and creates the framework for long-term development; it also contributes significantly to total economic growth. To reduce the environmental effect of regenerative tourist growth, businesses in the tourism sector can embrace more ecologically friendly modes of production; reduce the quantity of wasted materials, energy, and trash; and enhance the rate at which they recycle rubbish. Furthermore, it is critical to educate regenerative tourists about environmental concerns and to prohibit impolite conduct during tourism activities that are harmful to the natural environment. Tourism has a huge impact on the

natural environment in which it operates, as well as a substantial reliance on that natural ecosystem [76]. Tourism in the area may also contribute to the preservation of natural resources and the enhancement of the region's attractiveness [80].

Working in the regenerative tourism business requires one to create an atmosphere in which everyone engaged, including tourists and the surrounding environment, benefits. Moreover, the proceeds from tourism should be utilized to preserve natural regions, ensuring the long-term survival of resources [81]. Furthermore, locals should execute an environmental plan in order to provide guests with fresh green fruits and vegetables remove any processing connections and enhance the natural taste of the food while providing a delightful meal [82].

Maintaining and improving environmental protection is a critical component of any strategy for attaining long-term development. Furthermore, an understanding of relevant laws, rules, and policies, as well as a willingness to enhance the quality of their environmental protection is required. This is because environmental education teaches individuals how the human–land interaction system works, and as a consequence, people are more willing to care for the natural environment [83]. Poisonous gas emissions into the atmosphere, as well as the dumping of solid or liquid wastes in the land, air, or water, eventually end in environmental degradation and the loss of natural plant and animal habitats [84]. Road transportation is responsible for about 74% of global transportation emissions, as well as 20% of total greenhouse gas emissions. Road building may be damaging to animals because it destroys flora and soils, reduces accessible water sources, and reduces the agricultural area [85].

Large buildings (that obstruct views), competing architectural styles (that appear out of place), noise pollution from airplanes and vehicles, erosion, and graffiti (paint on walls) are examples of environmental degradation [86]. In general, excessive exploitation, haphazard planning, environmental harm, and energy consumption would almost likely result from a lack of environmental awareness and governance competency in the early stages of urban tourist growth. This is because urban tourist development causes overexploitation, environmental harm, and increased energy consumption [15]. The research's two more hypotheses are as follows:

**H6.** *CPECID has positive and negative environmental impact.*

**H7.** *Environmental impact has a positive relationship with tourism development.*

### 2.6. The Mediating Role of Economic Benefit, and Socio-Cultural and Environmental Impact

The initial hypotheses, which range from H2 to H7, are provided in order to investigate the connection between the economic benefits, socio-cultural impact, and environmental impact's antecedent factor (such as the construction of CPEC infrastructure) and the result variable (such as tourism development). In the current investigation, the hierarchy of the effect model was used since it was based on these ideas. Economic benefits, socio-cultural impact, and environmental impact all represent the local residents' internal evaluation process (the cognitive component), which in turn leads to the residents' affective responses to tourism development. This concept is based on the idea that residents' affective responses to tourism development are influenced by their cognitive responses. In other words, the construction of the CPEC infrastructure is not expected to influence the growth of regenerative tourism provided that local communities do not perceive any economic advantages, socio-cultural impacts, or environmental impacts from the project.

The growth of regenerative tourism is anticipated to benefit from the construction of CPEC infrastructure, as stated in Hypothesis 1. In addition, although the local population is evaluating the CPECID, it may not have a real concept of the growth of tourism in the local region. As a consequence of this, it may be hesitant to completely declare an increase in the local tourist market. However, if residents begin to perceive the economic benefits, socio-cultural impacts, and environmental impacts of CPECID, for example, by watching a

news channel or hearing positive word-of-mouth, they may still want to embrace tourism development. As a result, it is not unreasonable to consider the economic benefits, socio-cultural impact, and environmental impact of CPECID to be the mediators. In consideration of the explanations that follow below, the following presumptions are made:

**H8.** *(a) Economic benefits, (b) socio-cultural impact, and (c) environmental impact positively mediate between CPECID and tourism development.*

### 2.7. Conceptual Model

Figure 2 depicts the hypothetical model. The literature review is used to guide the selection of each component of the model. According to previous research, residents' views of the impacts of tourism have a role in influencing their degree of support for future tourism. The above-mentioned causal links between citizen support and the results of tourism are instances of tourist development theory. The three areas into which this study's hypothetical model categorizes the impacts that are supposed to be induced by the rise of tourism are economic, socio-cultural, and environmental. The model analyzes the structural relationship that exists between the different variables of CPECID, perceived visitor effects, and tourism development support. In theory, each aspect of tourism's (i.e., economic benefits, socio-cultural impact, and environmental impact) is influenced by the impact of the CPECID, which in turn has an impact on the amount of support provided to tourist growth. Finally, we consider the economic benefits, socio-cultural impact (benefit and cost), and environmental impact (benefit and cost) of CPECID as the mediators to predict the overall perception towards tourism development.

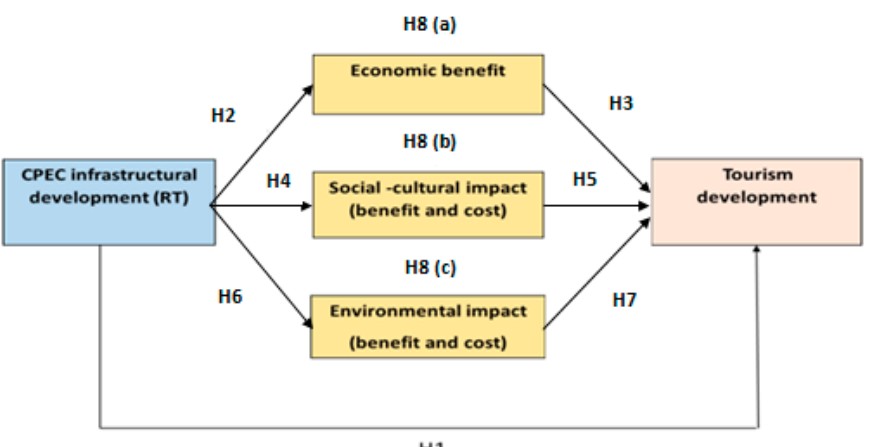

**Figure 2.** Conceptual model.

## 3. Methodology

### 3.1. Measures

The scale devised by [87] was used to generate the questions used to assess CPEC infrastructure progress. Similarly, the questionnaire used to assess the socio-cultural benefits and costs of tourism is employed through [25,60]. Likewise, the questionnaire used to quantify economic benefits is utilized through [25,60]. The same questionnaire used to assess environmental impact is used in the following study by [60,88–90]. Lastly, the questionnaire used to assess tourist development was utilized in the following study conducted by [60,91]. Cronbach's Alpha was higher than 0.70 for all measurements. For all factors, a five-point Likert scale answer choice was offered, with 1 indicating strongly disagree and 5 indicating strongly agree.

### 3.2. Participants and Procedures

Our survey questionnaire was divided into three sections. The first section comprised a summary of the study's goal, a request to complete the questionnaire, and assurances

of data protection and anonymity. The second segment includes questions about the respondents' demographics, such as age, gender, dwelling location, and education. The third section of the questionnaire contains questions on China–Pakistan economic corridor infrastructure development (CPECID), economic benefit, sociocultural impact (benefits and costs), environmental impact (benefits and costs), and tourist development.

Inhabitants (or residents) of GB participated in the survey that was used to gather the data using quantitative research technique. Quantitative research takes a logical approach that prioritizes data collection using standardized surveys [20,92–98]. With the help of a reputable panel company (businesses that pair online survey takers with the survey's intended audience for a charge per full answer) based in Islamabad (i.e., the capital of Pakistan), we were able to gather data from the broad Gilgit Baltistan population using Google forms and paper surveys.

The cover letter of the survey questionnaire included information on the benefits and objectives of the research. In addition, respondents were assured in the cover letter that their responses would remain confidential. In terms of the overall representativeness of the samples, the role of panel companies proved effective. It was discovered that the majority of respondents had a vested interest in tourist growth in GB and were not simply representative of the local community. This supports the researchers' opinion that the CPEC project has substantial popular support and ownership among Pakistanis [24,26,99].

There were 400 responses, and 336 of them were complete and useful, for an 84% response rate. The selected sample size represents a 5.35% margin of error at a 95% confidence interval given that the population of GB is around 1.5 million. The acceptable margin of error, according to many reputable survey sites Pollfish, 2023; Voxco, 2023; and Zoho Survey, 2023) and researchers (e.g., Lyons and Hearne; cited in Trone Research and Consulting, 2023) often ranges between 4% and 8% at the 95% confidence level.

Table 1 shows the sample's demographic characteristics. Participants in the survey were 62.8% male and 37.2% female. Out of 336 respondents, 264 were between the ages of 20 and 40, 68 were under the age of 20, and just four were between the ages of 41 and 60. In total, 62 respondents completed intermediate, 215 completed a bachelor's degree, and 59 completed a Master's degree or above. There were 285 single replies, 50 married, and 1 divorced.

**Table 1.** Characteristics of respondents' sample.

| Variable | Frequency | Percentage |
|---|---|---|
| Gender | | |
|     Male | 211 | 62.8 |
|     Female | 125 | 37.2 |
| Age | | |
|     Below 20 years | 68 | 20.2 |
|     20–40 years | 264 | 78.6 |
|     41–60 years | 4 | 1.2 |
| Education | | |
|     Intermediate | 62 | 18.5 |
|     Bachelors | 215 | 64 |
|     Master's and above | 59 | 17.6 |
| Marital Status | | |
|     Single | 285 | 84.8 |
|     Married | 50 | 14.8 |
|     Divorced | 1 | 0.29 |

## 4. Results

### 4.1. Measurement Model

In SMARTPLS, the first step to proceed with the analysis of the data is to assess the measurement model. In the measurement model, the items were assessed for the construct validity and reliability, convergent validity, and internal consistency. In the first step,

factor loadings (FLs), average variance analysis (AVEs), Cronbach's alpha, and composite reliability (CR) were assessed to check the internal consistency of the measurement items. For the constructs, the criteria of the internal consistency were achieved, as is evident from the values recorded in Table 2. The values for the FLs were between 0.505 and 0.909. These values are supported by the measurement model given in Figure 3, thus fulfilling the threshold values. In the same way, values for AVEs were achieved more than 0.5. Likewise, values attained for CR and CBA were more the 0.7.

**Table 2.** Measurement model for the study.

| Items | FL | CA | CR | AVE |
|---|---|---|---|---|
| CPECID1 | 0.787 | | | |
| CPECID2 | 0.878 | | | |
| CPECID3 | 0.729 | | | |
| CPECID4 | 0.782 | 0.908 | 0.928 | 0.684 |
| SCI1 | 0.878 | | | |
| SCI2 | 0.733 | | | |
| SCI3 | 0.738 | | | |
| SCI4 | 0.834 | | | |
| SCI5 | 0.879 | | | |
| SCI6 | 0.877 | 0.932 | 0.946 | 0.745 |
| EB1 | 0.869 | | | |
| EB2 | 0.909 | | | |
| EB3 | 0.859 | | | |
| EB4 | 0.772 | 0.937 | 0.950 | 0.761 |
| EI1 | 0.580 | | | |
| EI2 | 0.707 | | | |
| EI3 | 0.548 | | | |
| EI4 | 0.700 | | | |
| EI5 | 0.691 | | | |
| EI6 | 0.557 | 0.910 | 0.931 | 0.693 |
| EI7 | 0.505 | | | |
| TD1 | 0.708 | | | |
| TD2 | 0.716 | | | |
| TD3 | 0.829 | 0.943 | 0.955 | 0.779 |
| TD4 | 0.712 | | | |

Notes: CR: composite reliability; AVE: average variance extracted; and CA: Cronbach's alpha.

For assessing the discriminant validity, the FornellLarcker and Hetrotrait and Monotrait Ration (HTMT) tests was applied. Table 3 gives the values of the square root of the AVE in the diagonal cell, which needs to be higher, as per the FornellLarcker criterion to establish and check discriminant validity. Additionally, according to HTMT, all of the correlation values were less than 0.9, which achieved the recommendation of [100].

**Table 3.** Discriminant validity—FornellLarcker and HTMT.

| Constructs | HTMT CPECID | SCI | EB | EI | TD | BO | JP | OCB | POS | PS | WB |
|---|---|---|---|---|---|---|---|---|---|---|---|
| CPECID | 0.827 | | | | | | | | | | |
| SCI | 0.059 | 0.863 | | | | 0.089 | | | | | |
| EB | 0.017 | 0.205 | 0.845 | | | 0.081 | 0.219 | | | | |
| EI | 0.233 | 0.047 | 0.163 | 0.873 | | 0.230 | 0.078 | 0.174 | | | |
| TD | 0.521 | 0.140 | 0.088 | 0.412 | 0.832 | 0.549 | 0.147 | 0.099 | 0.441 | | |

The off-diagonal values are the correlations between latent variables, and the diagonal is the square root of AVE.

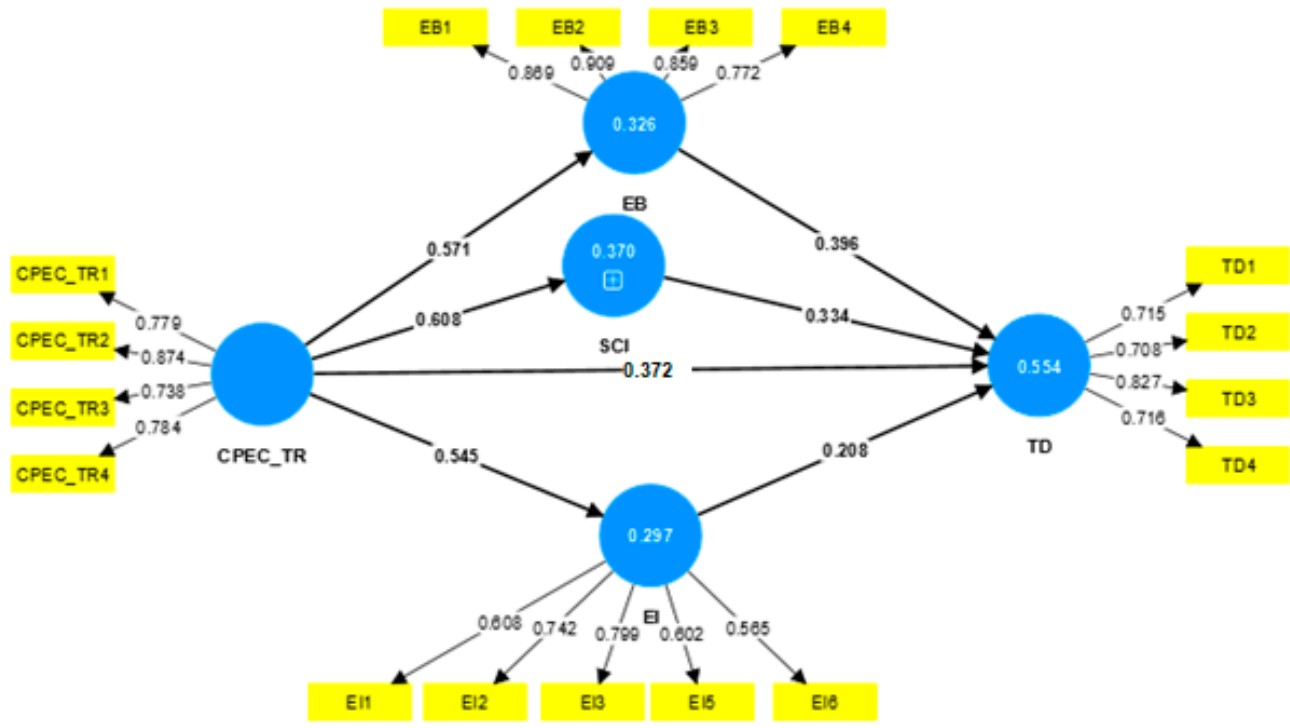

**Figure 3.** Measurement Model for the study.

### 4.2. Assessment of the Structural Model

After the assessment of the measurement model, we check the collinearity through inner VIF, the $R^2$ values, the effect size ($f^2$), and the predictive relevance ($Q^2$) in the structural model. All of the recommended values of $R^2$, $F^2$, $Q^2$, and inner VIF were achieved and have been presented in Table 4. Then, we proceed to observe the proposed hypotheses results.

**Table 4.** Assessment of the structural model.

| | Endogenous Variables | R Square | | R Square Adjusted | | |
|---|---|---|---|---|---|---|
| R-Square | EI | 0.118 | | 0.111 | | 0.26: Substantial, 0.13: Moderate, 0.02: Weak (Cohen 1989) [101] |
| | SCI | 0.164 | | 0.151 | | |
| | EB | 0.515 | | 0.511 | | |
| | TD | 0.241 | | 0.235 | | |
| **Effect Size (F-Square)** | Exogenous Variables | BO | JP | OCB | PS | 0.35: Large, 0.15: Medium effect, 0.02: Small effect (Cohen 1989) [101] |
| | CPEC_TR | | 0.018 | | | |
| **Collinearity (Inner VIF)** | Exogenous Variables | EI | SCI | EB | EI | |
| | EI | | 1.404 | | | VIF ≤ 5.0 (Hair et al. 2017) [102] |
| | SCI | | 2.106 | | | |
| | EB | 1.006 | 1.249 | 1.006 | 1.006 | |
| | TD | | 1.660 | | | |
| | EI | 1.005 | 1.870 | 1.005 | 1.005 | |

**Table 4.** *Cont.*

| | WB | CCR | CCC | |
|---|---|---|---|---|
| | EI | 0.074 | 0.563 | Value larger than 0 indicates predictive relevance (Hair et al. 2017) [102] |
| **Predictive Relevance (Q-Square)** | SCI | 0.117 | 0.637 | |
| | EB | 0.363 | 0.596 | |
| | TD | 0.162 | 0.571 | |

CCC = construct cross-validated communality, CCR = construct cross-validated redundancy.

For examining the proposed hypotheses, a 5000-resample bootstrapping procedure was used in Smart-PLS. The results are presented in Table 5. According to the results, all hypotheses were supported. For all hypotheses, the Beta values (β) were attained more than 0.10, which means that they play a significant role in TD, both directly and indirectly. Similarly, the *p*-values and T-values for all of the relationships were achieved under its threshold, i.e., ($p \leq 0.05$) and (T $\geq$ 1.96). The details are presented in Table 5, and the findings are also supported by the structural model, given in Figure 4.

**Table 5.** Hypotheses testing result.

| Direct Hypothesis | β-Values | M | STDEV | T-Values | *p*-Values |
|---|---|---|---|---|---|
| CPEC_TR → EB | 0.571 | 0.580 | 0.088 | 6.458 | 0.000 |
| CPEC_TR → EI | 0.545 | 0.555 | 0.104 | 5.230 | 0.000 |
| CPEC_TR → SCI | 0.608 | 0.614 | 0.077 | 7.891 | 0.000 |
| CPEC_TR → TD | 0.543 | 0.557 | 0.087 | 6.262 | 0.000 |
| EB → TD | 0.396 | 0.412 | 0.094 | 4.234 | 0.000 |
| EI → TD | 0.208 | 0.208 | 0.112 | 1.856 | 0.004 |
| SCI → TD | 0.334 | 0.334 | 0.103 | 3.235 | 0.001 |
| CPEC_TR → EB → TD | 0.226 | 0.239 | 0.068 | 3.330 | 0.001 |
| CPEC_TR → SCI → TD | 0.203 | 0.203 | 0.065 | 3.116 | 0.002 |
| CPEC_TR → EI → TD | 0.113 | 0.114 | 0.066 | 3.711 | 0.007 |

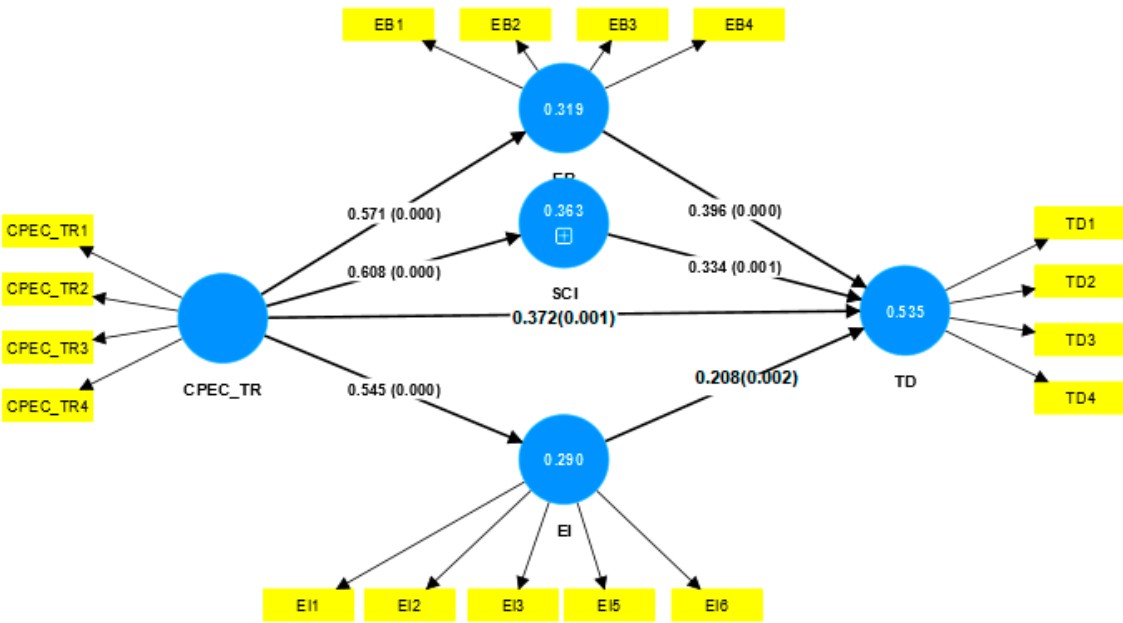

**Figure 4.** Structural model with inner model path coefficient values and *p*-values.

## 5. Discussions and Conclusions

The goal of this study was to discover how the CPEC road infrastructure has influenced regenerative tourist growth by investigating the roles of economic benefits, social-

cultural impact, and environmental impact as mediators. Because the development of GB is projected to have far-reaching consequences for both the local and global economies, this analysis has focused only on the GB tourism market sector. The conceptual model employed in this research was substantially based on past studies that had already covered all of the topics being explored here, and it was validated for correctness by native GB residents.

The empirical data support the first hypothesis that CPEC infrastructure development (CPECID) has a favorable effect on the regenerative tourist development in GB. This result is consistent with past findings that infrastructure expansion encourages tourist growth [3,4,9,14,103]. Therefore, the research confirms that the expansion of CPEC roads and transportation infrastructure has had a favorable influence on regenerative tourism in GB, Pakistan.

The empirical data also support the following six hypotheses (H2–H7): CPEC infrastructure development (CPECID) has a positive impact on economic benefits, social-cultural impact, and environmental impact; eventually, economic benefits, social-cultural impact, and environmental impact have a positive impact on tourist development in GB. These findings are consistent with the insights that citizens' views of the positive economic advantages of tourism inspire and compel them to support tourism expansion in their community [66,67]. Moreover, the construction of local area infrastructure and subsequent encounters with tourists has a direct influence on community views of tourism's consequences, especially among those who benefit the most from tourism [59–62]. Furthermore, the influence of CPECID on locals' perceptions of the impacts of regenerative tourism and their support for the industry's expansion is favorable in terms of environmental factors [19].

The use of the hierarchy of impact model based on the specified prepositions is the most notable component of the current research. In this method, the internal process of the voters, i.e., the cognitive component, was represented by economic benefits, socio-cultural (benefits and costs), and environmental (benefits and costs). This process stemmed from local inhabitants' assessments of CPECID, which led to their emotional reactions, i.e., their impression of regenerative tourist development. According to the findings of this research, economic advantages, socio-cultural (benefits and costs), and environmental (benefits and costs) benefits mediate the relationship between CPECID and tourist development in GB. Previous studies indicated that owing to substantial environmental concerns, people's perceptions of tourist activities and their environmental impact are often unfavorable [5,104,105]. However, the findings of this research suggest that environmental impact has a favorable partial mediating function. This could be due to social media and the government highlighting the positive environmental impacts of CPECID in terms of improved quality of life, the construction of socially responsible regenerative tourist destination points and resorts, the placement of socially responsible messages and bins at various locations, and the development of modern facilities.

This study's findings are corroborated by research by [106] which claims that the good implementation of environmental protection legislation may produce economic activity and encourage tourism. Because of these favorable perceptions, the detrimental environmental effect is minimal. As a result, the inhabitants of GB are eager to welcome more regenerative tourists to their region. A few studies have shown that community satisfaction is a strong predictor of community support for tourism [24,99]. People are satisfied with the steps made by the government and local governments to prevent pollution along the CPEC corridor.

In terms of the socio-cultural influence of CPECID on regenerative tourist development, residents are enthusiastic about supporting local community events and festivals. According to the general people of the GB, regenerative tourism will aid in the promotion of their culture and traditions around the globe, indicating support for regional tourist operations. According to the findings, tourism is already having a beneficial economic influence on the lives of the people of the GB. Regenerative tourism has created job and

investment prospects in the region. The construction of CPEC infrastructure has provided commercial prospects for locals and small enterprises. These findings are backed by a few other academics, which show that the regenerative tourist sector makes a big contribution to any country's foreign currency reserves, supplying a large percentage of the population with direct and indirect job possibilities. Furthermore, it may encourage a country's arts and crafts to preserve nature's beauty, cultural legacy, and history [107]. Regenerative tourism is currently seen as a new source of employment generation. Tourism boosts consumption and accelerates economic development [104].

## 6. Limitation and Future Research

There are some constraints, and as a result, there must be further study, as well as the potential for additional research in the near future on aspects of the subject that have not been investigated yet. First, this project only had a small number of samples because of a lack of time and money. Future researchers can gather large numbers of data to make the topic more general. Our main goal is to help tourism grow, especially in GB. The focus of future research should be on gathering information from Cashgar to Gawadar port (i.e., other important territories of Pakistan in terms of execution of CPEC). Additionally, research should be carried out on places to visit along CPECID in the winter and summer. In addition to infrastructure development, future research may look at other latent variables, such as personal income, skills, development, residents' reduction in poverty and increase in prosperity, wildlife migration, and the environmental effects of CPEC on residents, and how they help tourism development.

**Author Contributions:** Conceptualization, M.Z.; Methodology, S.B.; Software, J.A.T.; Validation, J.A.T.; Formal analysis, J.A.T.; Investigation, J.A.T.; Resources, M.H.; Data curation, J.A.T.; Writing—original draft, S.B.; Writing—review & editing, S.M.A.Y.; Supervision, S.B.; Project administration, M.Z. and S.B.; Funding acquisition, S.M.A.Y. All authors have read and agreed to the published version of the manuscript.

**Funding:** The APC of this research is sponsored by Scientific Articles Support Fund (Fondo Apoyo Artículos Científicos), Tecnológico de Monterrey, Monterrey, Mexico.

**Institutional Review Board Statement:** Not applicable.

**Informed Consent Statement:** Not applicable.

**Data Availability Statement:** Data available on request due to privacy/ethical restrictions.

**Conflicts of Interest:** The authors declare no conflict of interest.

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
