# Peer review of "The Mediating Roles of Economic, Socio-Cultural, and Environmental Factors to Predict Tourism Market Development by Means of Regenerative Travel: An Infrastructural Perspective of China–Pakistan Economic Corridor (CPEC)"

_sustainability, doi:10.3390/su15065025_

Round 1

Reviewer 1 Report

An interesting and, generally, well-written paper - some very small typos, so a careful edit required.  The only comments I would make are, firstly, a definition of Regenerative Travel would be useful and, secondly, section 3.2 on participants does not make it clear who the respondents were in terms of the sampling.  Yes, it states that a panel company was used, but in terms of the sample are they representative (and in what way)? There's very little justification/explanation on whether the respondents have a vested interest in tourism development in the areas, or whether they are just representative of the local population. 

Author Response

Respected Review, 

I highly appreciate your comments. 

Here is the summary of adjustments made as per your kind suggestions:

  1. The only comments I would make are, firstly, a definition of Regenerative Travel would be useful.

Thanks for the suggestions: The required changes are incorporated. Please see first two paragraphs of section 1.2. Please see the green highlights. 

  1. Section 3.2 on participants does not make it clear who the respondents were in terms of the sampling.

Thanks for suggestions:

“Inhabitants (or residents) of GB participated in the survey that was used to gather the data using quantitative research technique.”

We have adjusted this statement in the section 3.2. Please see the green highlights. 

  1. Yes, it states that a panel company was used, but in terms of the sample are they representative (and in what way)?

Thanks for the suggestions: The required changes are incorporated in section 3.2. Please see the green highlights. 

  1. There's very little justification/explanation on whether the respondents have a vested interest in tourism development in the areas, or whether they are just representative of the local population.

Thanks for the suggestions: The required changes are incorporated in section 3.2. Please see the green highlights. 

Reviewer 2 Report

The article deals with an interesting topic in an interesting territory.

In relation to the article I make the following recommendations:

I think it is necessary to explain the concept of regenerative tourism as an approach that goes beyond sustainable tourism. In this regard, other bibliographical references that analyse this issue can be used. 

Although the effects of the CPEC are described, it is not explained what the project consists of. I also consider it important to include a map of the location of GB and some data on the territory. The authors do not explain its interest as a tourist destination or its potentialities. In tourism, the territory is fundamental and this work does not provide any data. 

The authors should justify more clearly the relationship between regenerative tourism and social exchange theory.

Finally, the authors should clarify the level of knowledge that respondents had about the CPEC project. Did they include any questions about it? 

Thanks

Author Response

Respected Reviewer, 

Thanks for your suggestions. 

We have made the following adjustments based on your kind suggestions:

  1. I think it is necessary to explain the concept of regenerative tourism as an approach that goes beyond sustainable tourism. In this regard, other bibliographical references that analyse this issue can be used.

Thanks for the suggestions: The required changes are incorporated. Please see first two paragraphs of section 1.2. Please see the green highlights. 

  1. Although the effects of the CPEC are described, it is not explained what the project consists of.

Thanks for the suggestions: The required changes are incorporated in section 1. Please see the green highlights. 

  1. I also consider it important to include a map of the location of GB and some data on the territory.

Thanks for the suggestions: The required changes are incorporated in section 1.1. Please see the green highlights. 

  1. The authors do not explain its interest as a tourist destination or its potentialities. In tourism, the territory is fundamental and this work does not provide any data.

Thanks for the suggestions: The required changes are incorporated in section 1.1. Please see the green highlights. 

  1. The authors should justify more clearly the relationship between regenerative tourism and social exchange theory.

Thanks for the suggestions. We believed that we already justified more clearly the relationship between regenerative tourism and social exchange theory. However, based on your indications, we added some more clarifications to our point of view. Please see the adjustments in section 2.1 with green highlights. 

  1. Finally, the authors should clarify the level of knowledge that respondents had about the CPEC project. Did they include any questions about it?

Thanks for the suggestions: The required changes are incorporated in section 3.2. Please see the green highlights. 

Reviewer 3 Report

I have the following comments:

- the size of the sample (336 respondents) and its representativeness should be considered,

- Based on the data on the given sample, it is not possible to make a conclusion that would confirm or reject the hypotheses. The authors also admit this fact. The manuscript should provide sample data: Confidence Level, Margin of Error, Population Proportion, Population Size, required sample size, and actual sample size. The representativeness of the selection of respondents is questionable. At the same time, add data about the population in the text (representativeness of the selection) or target the research only to people who work in the tourism industry.

- line 505 contains the following typo: Fornell Larker - Fornell Larcker,

- line 529 contains the following typo: the details – The details

- consistency of references in the text with references at the end of the text (for example, Henseler et al. (2015), Cohen (1989), Hair et al. (2017) are not in the references).

It depends on the opinions of the editor and other reviewers whether the conclusions (based on this sample) are statistically relevant.

English language and style are fine. Minor spell check is required.

Author Response

Respected Reviewer, 

Thanks for your kind suggestions. 

Based on your comments, we have made the following adjustments into our manuscript:

  1. The size of the sample (336 respondents) and its representativeness should be considered. Based on the data on the given sample, it is not possible to make a conclusion that would confirm or reject the hypotheses. The authors also admit this fact.

Thanks for your suggestions. However, it is requested to understand that although we admit that the current project have a small number of samples (i.e., because of a lack of time and money), we did not claimed that they are not generalize able. The future researchers can gather large amounts of data to make the topic more general.

The survey data validity is already proven in the research to make a conclusion that would confirm or reject the hypotheses. Please refer to the following link for more information related to acceptable range of margin of error and sample size: https://www.troneresearch.com/blog/sample-size-requirements-reliable-study#:~:text=In%20general%2C%20the%20precision%20of,the%20point%20of%20diminishing%20returns.

  1. The manuscript should provide sample data: Confidence Level, Margin of Error, Population Proportion, Population Size, required sample size, and actual sample size.

Thanks for the suggestions: The required changes are incorporated in section 3.2. Please see the green highlights. 

  1. The representativeness of the selection of respondents is questionable. At the same time, add data about the population in the text (representativeness of the selection) or target the research only to people who work in the tourism industry.

Thanks for the suggestions: The required changes are incorporated in section 3.2. Please see the green highlights. 

  1. Line 505 contains the following typo: Fornell Larker - Fornell Larcker

Thanks for the suggestions: Adjusted.

  1. Line 529 contains the following typo: the details – The details.

Thanks for the suggestions: Adjusted.

  1. Consistency of references in the text with references at the end of the text (for example, Henseler et al. (2015), Cohen (1989), Hair et al. (2017) is not in the references).

Thanks for the suggestions: Adjusted.

Round 2

Reviewer 2 Report

Thank you for your reply

Author Response

Dear Reviewer, 

Your suggests are highly worthy to refine our manuscript. In fact, we found that the manuscript is more refined now. 

I hope this will be good enough to be fully accepted. 

Regards, 
